# OpenReview forum: "Deception in Large Language Models: An Audit Game–Theoretic Analysis"
_ICLR.cc/2026/Conference — Submitted to ICLR 2026_

### Official Review · Reviewer_bFTa · 2025-11-01

**Soundness:** 3
**Presentation:** 2
**Contribution:** 3
**Rating:** 6
**Confidence:** 3

**Summary:**

The paper models llm deception using an audit-game framework, where deception intensity is treated as a continuous variable and audit probability influences equilibrium behaviour.
Through four experimental phases (no audit, uncertain audit, quantified audit and pre-audit screening), the authors show that reasoning-oriented models tend to reduce deceptive behaviour when audits are made explicit.

**Strengths:**

1. The paper models deception as a continuous decision variable within an audit-game framework, providing an interpretable and theoretically grounded way to analyse strategic LLM behaviour.
2. The four-phase setup (no audit, uncertain audit, quantified audit, pre-audit) offers a systematic framework for studying how models respond to varying audit pressure.
3. The inclusion of a pre-audit “deception confidence” component adds a practical element to the otherwise theoretical analysis and helps link the framework to potential real-world applications.

**Weaknesses:**

1. The paper would benefit from a dedicated related-works section to better position its contribution within recent research on llm deception, oversight and audit-game frameworks.
2. Some derivation steps between Equations (12) and (14) are omitted. Including a short walkthrough in the appendix showing how the best-response conditions lead to the stated equilibrium expressions would make the theoretical section easier to interpret.
3. The mapping between theoretical quantities (such as deception intensity and auditor cost) and empirical measures is somewhat abstract. Clarifying how these are estimated from model outputs would strengthen the link between theory and experiment.
4. Figure 1, while visually appealing, adds limited analytical value and could be simplified or replaced by a clearer schematic of the audit-game structure.
5. Figure 2a could benefit from error bars or other uncertainty indicators to convey the reliability of the reported averages.

**Questions:**

1. How were sample sizes and repetitions determined for each audit phase?
2. How is “deception intensity” defined or inferred from model outputs, and how does it relate to the formal variable x in the theoretical model?
3. How sensitive are the findings of the pre-audit “deception confidence” mechanism to different parameter settings or thresholds?

---

> ### Author Response · Authors · 2025-11-27
>
> Global Response to Area Chair:
> We thank the Area Chair and reviewers for their constructive feedback. We have addressed all concerns, resulting in a more rigorous manuscript. Below, we summarize the core contributions of our work and the major revisions made during the rebuttal phase.
> Summary of our work: This paper focus on the acute safety challenge of deception in LLMs. We introduce Audit Game Theory to quantitatively model the adversarial dynamic between a utility-maximizing LLM agent and a resource-constrained auditor. Our key contributions are:
> - We apply audit game theory to analyze LLM deception, elevating the analysis from binary deception detection to a continuous, game-theoretic quantification of deception intensity.
> - We empirically reveal a divergence across model types: reasoning models adjust deception rationally only under explicit audit risks, whereas non-reasoning models remain largely insensitive to enforcement, showing that vague or ambiguous audit threats offer little deterrence. LLMs exist a self-assessment of output when deceive.
> - We propose and empirically verify a pre-audit mechanism that uses model-generated self-assessment to reduce deceptive actions while simultaneously lowering auditing costs.
> Summary of Major Revisions
> We have systematically revised the manuscript across three core modules: Theoretical Formulation (formalizing definitions and derivations), Experimental (operationalizing prompts, metrics mapping, and data samples), and Presentation (expanding related work and refining framework visualizations)
> We believe these revisions fully resolve the reviewers' concerns and strengthen the paper's contribution to the field of AI Safety.
>
> To Dear reviewer:
> We sincerely thank the reviewer for the positive assessment and the constructive comments. We have significantly revised the paper as follow.
> Response to Weakness 1 (Dedicated Related-works Section): We fully agree with the reviewer that a comprehensive discussion of related work is essential to properly position our contribution. However, due to the strict page limits, we were unable to include a full separate section in the main text. We have adopted a two-fold strategy in the revision:
> - Literature review: We have explicitly structured the literature review within the Introduction to clearly contrast our work with existing studies on LLM deception.
> - Extended Related Work in Appendix: We have added a dedicated "Related Work" section in Appendix. This section provides a detailed, systematic review of recent research on LLM strategic behavior, AI safety, and game-theoretic safety frameworks.
> Response to Weakness 2 (Derivation steps):
> To improve the interpretability of the theoretical section, we have added step-by-step explanations of the formulas. Additionally, following the suggestions of other reviewers, we revised the placement of the formulas and expanded the detailed explanation of our framework.
> Response to Weakness 3 (Mapping theory to empirics): We clarify that the deception intensity $x$ corresponds to the severity of the fabricated claim items (ordered from Item 1 to Item 4 based on increasing monetary value and decreasing relevance). We also found that LLMs indeed engage in self-evaluation when generating claims.
> Response to Weakness 4 (Figure 1): We agree that the original Fig. 1 prioritized aesthetics over clarity. We have redesigned Fig. 1 to be a functional schematic of the Audit Game structure.
> Response to Weakness 5 (Error bars and reliability): We appreciate this suggestion regarding the statistical rigor of Figure 2a. While we ideally would have performed large-scale repetitions for every data point, we were constrained by the high inference costs associated with the commercial sota LLMs evaluated. However, to ensure reliability, we did conduct repeated trials (N>2) on some LLMs. In these validation runs, we observed that the variance was negligible (very small error bars), indicating that the models' strategic behavior is highly stable under our prompt conditions.
> Response to Question 1 (Sample sizes): e generated 121 unique insurance claim scenarios, derived from a seed dataset to cover diverse accident types. Each scenario was tested across 8 models and 4 phases, resulting in a total of over 3,100+ interaction traces.
> Response to Question 2 (Deception intensity definition): In our theoretical model, $x$ is a continuous variable representing the magnitude of rule violation. To clarify this, we also modified our section of problem formulation.
> Response to Question 3 (Sensitivity and threshold): Our selection of the threshold $\delta$ was empirically calibrated based on the confidence score distributions observed in Phases A-C. we explicitly included this threshold and judgment criteria in the Pre-auditor's system prompt (see Appendix) to ensure consistent evaluation. Thus, the reported results reflect a data-driven balance point tailored to the model's behavioral baseline.

---

### Official Review · Reviewer_CttK · 2025-11-06

**Soundness:** 2
**Presentation:** 1
**Contribution:** 2
**Rating:** 2
**Confidence:** 4

**Summary:**

This paper claims to investigate "deception" in language models, both theoretically and empirically. It introduces an audit game and studies optimal behavior under some assumptions. Then it describes experiments in which the game is encoded into natural language for prompting language models and describes qualitative and quantitative results. It argues that the new framework it presents provides a foundation "for understanding, evaluating and governing strategic behavior in LLMs."

**Strengths:**

Studying deception in language models is an important area of research. The audit game setup seems novel.

**Weaknesses:**

The paper does not define many essential concepts, including "deception" itself. It does not provide enough detail to properly understand what is going on. Modeling decisions are not justified. Experiment details are unclear. Many of the claims seem to be insufficiently supported.

**Questions:**

I will index my questions by line number.

```
11. "high-risk settings" like what?
13. audit game theory -- what is this?
14. "deception is modeled as a continuous variable" -- why is this a reasonable model?
    and also: it's only kind of continuous; discrete choice, then continuous "intensity" of deception.
20. "deceive" -- what do you mean by deception? how do you know if models deceive?
22. "deception confidence" -- what is this?
23. "compress the feasible deception space" -- what does this mean?
    "mitigates" -- quantify? define?
24. "reduces the economic burden of auditing" -- quantify? define?
26. "foundations" -- in what ways could this allow someone to build on these foundations?

41. "deception [...] therefore poses higher risks" -- why?
63. "choose", "deceive" -- what do these mean?

85. I do not know what the paper is actually claiming at this point.
    I don't understand what the authors mean by "deception".

105. where did this equation (1) come from?
108. "When deception is detected" -- do you just mean the deception action?
111. Eq (2) -- why are you modeling it this way? why is this reasonable?
114. "more deceptions require greater investigatory resources" -- is this true?
     what _is_ deception?

201. '"open world" setting' -- what does this mean?
202. "deceptive strategy" -- what makes this _deception_?
     if you renamed the T/D actions the result would be the same for T, no?

212. "This theoretically proves [...]" -- for this one very specific game...
     what does it this say in general?
215. "It is possible [...]" -- _does_ it exist? is it guaranteed to exist?
     how do we know it's possible?

218. "consciously aware" -- what does this mean?
221. "We assume [...] it internally maintains an implicit assessment of its own
     degree of deception or truthfulness." -- why is it safe to assume this?
     what if it is just bullshitting?

259. "The cost C_pre is assumed to be much lower than C_a"
     can you make a list of all the assumptions? it's hard to keep track!

293. "is told that auditing exists" -- not clear
294. "is provided with" -- how?
299. "applies a grading rule" -- where does the rule come from? how is it applied?
300. "claims with C_B below a specified threshold"
     where does the confidence score C_B come from?
290--301. Not enough detail to follow what's going on here

324. error bars? significance?
329 & 345. The GPT-o4 numbers don't match.

375. "higher risk items" -- what does this mean?
376. "this shows LLM deception involves self-assessment rather than randomness."
     this is a big claim!

442. Fig 4. -- how did you get these numbers?
452. "detection confusion matrix" -- what is this?
464. "compresses the feasible deception strategy space"
     what does this _mean_?
484. "solid [...] foundation for engineering safer LLM systems."
     how would you do it?
```

---

> ### Author Response · Authors · 2025-11-27
>
> Dear Area Chair:
> We thank the Area Chair and reviewers for their constructive feedback. We have addressed all concerns, resulting in a more rigorous manuscript. Below, we summarize the contributions of our work and the major revisions made during the rebuttal phase.
> Summary of our work: This paper focus on the acute safety challenge of deception in LLMs. We introduce Audit Game Theory to quantitatively model the adversarial dynamic between a utility-maximizing LLM agent and a resource-constrained auditor. Our key contributions are:
> - We apply audit game theory to analyze LLM deception, elevating the analysis from binary deception detection to a continuous, game-theoretic quantification of deception intensity.
> - We empirically reveal a divergence across model types: reasoning models adjust deception rationally only under explicit audit risks, whereas non-reasoning models remain largely insensitive to enforcement, showing that vague or ambiguous audit threats offer little deterrence. LLMs exist a self-assessment of output when deceive.
> - We propose and empirically verify a pre-audit mechanism that uses model-generated self-assessment to reduce deceptive actions while simultaneously lowering auditing costs.
> Summary of Major Revisions
> In response to the reviews, we have extensively revised the paper to improve clarity, theoretical grounding. We have revised the paper across multiple dimensions, including the formalization of key definitions and concepts, clarification of settings and assumptions, enrichment of experimental details, and clearer presentation of claims.
> We believe these revisions fully resolve the reviewers' concerns and strengthen the paper's contribution to the field of AI Safety.
>
> To Dear reviewer:
> We thank the reviewer for the detailed feedback. We have significantly revised the paper as follow.
> Formalization of Definitions & Concepts (Line11, 13, 20, 22, 23, 24, 41, 63, 85, 201, 202, 218, 259, 375, 452, 464)
> - Deception (Line20, 63, 85, 202): We now explicitly define deception not as a psychological state, but as a strategic policy choice (x > 0) where the agent systematically deviates from the rule-adhering policy (Truthfulness, x=0) to maximize utility.
> - Some definition (Line13, 22, 23, 24, 259, 452, 464): We added a concise definition in Abstract and Introduction. We clarified that "deception confidence" is the agent’s internal estimate of success, and "compressing feasible space" refers to rendering high-intensity deception strategies sub-optimal (leading to rejection). We also defined the "detection confusion matrix" as a binary classification task (Deceptive vs. Truthful).
> - Ambiguous term (Line11, 201, 375): We replaced vague terms like "high-risk settings" with specific examples and renamed "open-world" to " no audit" to avoid ambiguity.
> Clarify some settings and assumptions (Line14, 105, 111, 114, 221, 259, 442)
> - Continuous Deception & Equations (Line14, 105, 111): We clarify that while the decision to deceive is binary, the intensity (e.g., fraudulent amount) is continuous. We not only cite foundational work for our equations, but also add revise the definition of continuous deception.
> - Concern of "Bullshitting" not deception (Line221): We add new content about the different between deception and bullshitting. Our model posits the LLM as a rational agent to maximize utility. In our framework, maintaining an internal assessment of the chosen action is a computational prerequisite for utility maximization, not an assumption of consciousness.
> - Results clarify (Line442): we added new content to clarify how to obtain the result of Fig .4 in our manuscript. The claims used for evaluation were generated by the agent during Phase B and Phase C, and the corresponding classification results were produced by the LLM-based pre-audit mechanism.
> Experimental details (Line290-301, 300, 329, 345, 442)
> - Experiment setting (Line290-301): We replaced narrative descriptions ("is told") with specific operational steps. We now explain that parameters are injected via system prompts (full templates added to Appendix). We also detailed the "Pre-audit Mechanism" implementation.
> - Data Integrity (Line329, 345): In Fig. 2(a), we show the deception rates of different LLMs. However, the text on Line 345 refers to the refusal rate rather than the deception rate.
> Clarification of Claims (Line26, 212, 215, 376, 484)
> Foundations (Line26, 484): We rewrote the abstract and conclusion to specifically explain the meaningful of our work.
> Generalizability (Line212, 376): We toned down "theoretically proves" to "demonstrates within our quadratic utility framework." We clarify that "self-assessment" is inferred from the non-uniform drop in deception rates across relevance levels. By analysis the output of LLMs, they indeed have the process of self-assessment and trade off the risk and benefits.
> Threshold Existence (Line215): We have modified this rigorous term.

---

### Official Review · Reviewer_Ca4d · 2025-11-11

**Soundness:** 2
**Presentation:** 3
**Contribution:** 2
**Rating:** 4
**Confidence:** 2

**Summary:**

This paper introduces a game theoretic framework to explain deceptive behavior by LLMs in adversarial environments where deception is continuous and there is an auditor who can uncover deception, but at some cost. It sets up an idealized version of the problem and solves it exactly, then sets up experiments to evaluate how LLMs behave in the presence of different assumed auditor behaviors in the prompt. Overall, reasoning LLMs appear to become less deceptive when knowing that there is some probability that they will be audited, but not simply when knowing an auditor exists, whereas non-reasoning llms do not alter their behavior. Additionally, a pre-audit step involving checking with another LLM if the output from the agent is reasonable has good performance.

**Strengths:**

The framework described for deception is clear and well explained, and the experiments are straightforward. The impact of stated probabilities on deception level is strong for reasoning models, which validates the hypothesis of the paper.

**Weaknesses:**

Broadly, there is a disconnect between the theoretical and empirical sections of this paper. The theoretical section spends significant effort in establishing specific predictions on how much deception should occur under different auditing conditions, solving equations in a simple setting. These equations are then not referred to in the empirical section, where models appear to only occasionally be following the game theoretic framework described earlier. There are qualitative comparisons, but no quantitative comparison, even when the models do appear to be following the same qualitative model as the Nash equilibrium found earlier. Additionally, the x variable representing degree of deception does not appear to be measured or discussed in the empirical section.

In addition, Phase D feels unconnected from the rest of the work, and largely dependent on the domain in a way that the other parts of the experiment are not. (I.e., how good the pre-audit is is dependent on the domain in a more complicated way than a simple cost scalar like with the audit, and no exploration is made of the models trying to get around the pre-audit knowing some details about it.)

Finally, interpreting anything out of changes to a model's prompt is difficult as models can often substantially vary their behavior based on minor changes to a prompt, such as the formality of the language, "flattery", threats, etc. As such, any interpretation of model behavior needs to be caveated with the understanding that there are multiple possible interpretations of a change in behavior, as a single change might have multiple effects.

Paragraph/line level suggestions for improvement:

You should refer to a "Principal" or "Auditor" using one consistent name (I suggest Auditor, as it is more evocative of it's role).

Are Equations 6-8 necessary if you retreat to an LP formulation in equation 9?

Equation 14 appears to imply that as the cost of the fine increases (gamma goes up), the auditor would prefer to audit less (that is, x_indiff goes up). This appears to be because in the regime corresponding to this root, auditing becomes less effective relative to not auditing as the level of deception increases. This should be clearly explained and justified in the text, as it is relevant to understanding whether the model described in the paper makes qualitative sense.

96-97: If the agent only has two moves, where does deception intensity come in? Surely it has moves of the form T, D(x).

108-109: it should be made clear here that the auditor catching the agent means that it does not get the additional deception reward, but does get the base reward. This is encoded in equation 4 but should be flagged in the text earlier.

133-147: Equations 5-6, 8 and some surrounding context drop the superscript D for the utility under the deception assumption.

148: equation 9 I assume means actually U_L with no superscript; this should be defined somewhere. Equation 8 defines the optimal utility, not U_L.

155-157: why is R_0 relevant to the auditor?

164-165: equation 12 is two equations concatenated onto a single line

221: what is LLM-A?

229-232: C_B and delta are never defined

232: the pre-audit model is first introduced here, with it's last mention being in the introduction. It should be given a detailed explanation and motivation before equations are introduced in this section.

340-360: The first paragraph of Section 4.1 contains a few statements that are speculative based on the results presented, and should be flagged as such. (1) "implying that under uncertainty pressure, it adopts a “high-risk, high-reward” strategy rather than risk avoidance" could also be explained by a the fact that a reference to auditors in the prompt might lead it to "realize" deception is an option (2) " Over-all, the absence of disclosed audit parameters weakens both the calculability and perceptibility of deterrence, leading some models to interpret “auditing” as “low-frequency, avoidable spot checks,”which does not affect their marginal decisions" is more justified in presence of Phase 3 results but only for some models, not all of them. Without these results it might be the case that the model is just ignoring instructions regarding an auditor.

366-370: given how limited and scattered the Figure 2a results are, I do not think this analysis of Figure 2b really makes sense, for example, the non-reasoning models got more deceptive primarily because 4.1 did. Additionally, box-and-whisker plots should not be used for N=3 reasoning models.

371-377: I think this discussion could benefit from the table being of the relative differences between the two settings rather than the absolute numbers, as such, it is hard to tell what the percentages refer to, or if they are consistent across the various models. Some discussion about how relevance impacts deception would also be helpful for people unfamiliar with this domain.

430-431: reordering the models so reasoning ones appear adjacently would be helpful for the legibility of this figure. Additionally, the reasoning models should be specifically identified via some annotation on the figure (this should be done consistently throughout the paper)

445: the rho should be provided here too as it was provided above.

**Questions:**

See Weaknesses section for questions

---

> ### Author Response · Authors · 2025-11-27
>
> Dear Area Chair:
> We thank the Area Chair and reviewers for their constructive feedback. We have addressed all concerns, resulting in a more rigorous manuscript. Below, we summarize the contributions of our work and the major revisions made during the rebuttal phase.
> Summary of our work: This paper focus on the acute safety challenge of deception in LLMs. We introduce Audit Game Theory to quantitatively model the adversarial dynamic between a utility-maximizing LLM agent and a resource-constrained auditor. Our key contributions are:
> - We apply audit game theory to analyze LLM deception, elevating the analysis from binary deception detection to a continuous, game-theoretic quantification of deception intensity.
> - We empirically reveal a divergence across model types: reasoning models adjust deception rationally only under explicit audit risks, whereas non-reasoning models remain largely insensitive to enforcement, showing that vague or ambiguous audit threats offer little deterrence. LLMs exist a self-assessment of output when deceive.
> - We propose and empirically verify a pre-audit mechanism that uses model-generated self-assessment to reduce deceptive actions while simultaneously lowering auditing costs.
> Summary of Major Revisions
> We have revised the paper across multiple dimensions, including the Mathematical Rigor & Logic, Definitions & Conceptual Clarity, Experimental Interpretation, and visualization of manuscript.
> We believe these revisions fully resolve the reviewers' concerns and strengthen the paper's contribution to the field of AI Safety.
>
> To Dear reviewer:
> We thank the reviewer for the detailed feedback. We have significantly revised the paper as follow.
> Mathematical Formulations & Logic(Line 96, 108, 133, 148, 155, 164, Equation 14)
> - Strategy Space (Line 96): We clarified that the strategy is a continuous choice of intensity x, where T corresponds to $x=0$ and D corresponds to $x>0$.
> - Payoff Structure (Line 108, 155): We explicitly stated the payoff consequences (loss of $B(x)$, baseline payoff $R_0$) earlier in the text. We also clarified that while R_0 is not part of the auditor's utility, and modified our manuscript.
> - Notation Fixes (Line 133, 148, 164): We corrected the missing superscripts $U^D$, defined the non-superscripted $U_L$ as the overall expected utility, and fixed the Eq. 12.
> - Equation 14: We have clarified in the revised text that this result reflects the stability condition of the mixed-strategy equilibrium, not a failure of deterrence. This does not imply less policing. We have added a concise explanation of this dynamic to the text.
> Definitions & Notations(Line 221, 229, 232, 445)
> - Pre-audit Model (Line 229-232): As requested, we inserted a dedicated paragraph before the equations to motivate the pre-audit mechanism and formally define the confidence score $C_B$ and the rejection threshold $\delta$.
> - Terminology: We adopted the reviewer's suggestion to use "Auditor" consistently throughout the manuscript.
> - Undefined Terms: We defined LLM-A (the claimant agent) and provided the value for $\rho$ in the section of experiment.
> Experimental Interpretation & Visualization(Line 340-360, 366, 371, 430)
> Reducing Speculation (Line 340-360): We agree that the "high-risk, high-reward" claim was speculative. We have revised this section according to author’s suggestion.
> Table & Figures (Line 371, 430): Due to table-width constraints and layout considerations, we were unable to include additional columns without compromising readability. Instead, we have incorporated a dedicated discussion in the manuscript to clarify how “relevance’’ functions as a proxy for detectability and how it influences deceptive behavior across different models. We also clarified the relevance between different deception items. We reordered the models to group reasoning models together and added annotations in the figure.
> Regarding "Disconnect between Theory and Empirics": We respectfully disagree that there is a disconnect. we tested explicit audit probabilities of 5%, 10%, 20%, and 75%. Our experiment in Phase C quantitatively validates this.
> Regarding "Phase D Connection and Domain Specificity": Phase D is integral to the paper as it represents the solution (mechanism design) to the problem analyzed in Phases A-C. It’s base on the experiment results of Phase A-C. In Phase A-C, we demonstrated that LLMs have self-assessment of deception. Therefore, we can make use of this conjecture.
> Regarding "Prompt Sensitivity": Controlled Variables: We used a unified, rigid prompt template (see Appendix) where only the game parameters (e.g., the numerical value of $p$) were altered, while keeping the linguistic style, tone, and structure constant to minimize confounding artifacts like "flattery."  The fact that we observe consistent, monotonic behavioral shifts across multiple reasoning models and varying probabilities suggests that the signal of strategic reasoning is robust enough to transcend random prompt noise.

---

### Meta-Review · Area_Chair_mjdx · 2026-01-06

**Summary:**

This paper introduces a game theoretic framework to explain deceptive behavior by LLMs.
The authors proposed some mathematical modelings of costs and benefits of LLMs as well as auditors.
Then they derived optimal deceptive strategies for LLMs and auditing strategies for auditors.

The reviewers agreed that the paper is novel in some extent.
However, some essential concerns were raised regarding the rationale of the proposed cost/benefit modeling and the gap between the theoretical framework and experimental details.

**Reviewer Concerns:**

Cocerns regarding the rationale of the proposed cost/benefit modeling and the gap between the theoretical framework and experimental details remain unaddressed.
In particular, the modeling part is an essential component of the paper, and the lack of strong rationale for the modeling significantly weakens the contribution of the paper.

**Reviewer Scores:**

Reviewer Ca4d might have chance to increase the score from 4 to 5 if the concerns were addressed.
Reviewer CttK also might have chance to increase the score from 2 to 3 or 4 but it is less likely.
Reviewer bFTa is slightly positive and will remain the same at 6.

---

### Decision · Program_Chairs · 2026-01-26

Reject